# Virtual Environment Rehabilitation for Patients with Motor Neglect Trial (VERMONT): A Single-Center Randomized Controlled Feasibility Trial

**DOI:** 10.3390/brainsci11040464

**Published:** 2021-04-06

**Authors:** Elsje de Villiers, Thomas Stone, Nai-Wei Wang, Viswadeep Sarangi, Adar Pelah, Nicholas Shenker

**Affiliations:** 1Department of Physiotherapy, Addenbrooke’s Hospital, Hills Road, Cambridge CB2 0QQ, UK; elsje.devilliers@addenbrookes.nhs.uk; 2Department of Medical Physics, Addenbrooke’s Hospital, Hills Road, Cambridge CB2 0QQ, UK; thomas.stone@addenbrookes.nhs.uk; 3Hull York Medical School, John Hughlings Jackson Building, University Rd, Heslington, York YO10 5DD, UK; hynw4@hyms.ac.uk; 4Department of Electronic Engineering, University of York, York YO10 5DD, UK; v.sarangi@york.ac.uk (V.S.); adar.pelah@york.ac.uk (A.P.); 5York Biomedical Research Institute, University of York, York YO10 5DD, UK; 6Rheumatology Research Unit, Addenbrooke’s Hospital, Hills Road, Cambridge CB2 0QQ, UK; 7Department of Medicine, University of Cambridge, Hills Road, Cambridge CB2 0QQ, UK

**Keywords:** virtual environment, clinical trial, motor neglect, complex regional pain syndrome, sciatica, rehabilitation

## Abstract

Background: Motor neglect occurs in patients with chronic pain conditions. Virtual environments (VE) help rehabilitation through biofeedback and improving motivation. Aim: To assess the feasibility of a VE for patients with motor neglect with chronic pain. Methods: 10 subjects with chronic pain (Fibromyalgia, Sciatica, and Complex Regional Pain Syndrome) underwent a treadmill task three times per week for two weeks. Groups were randomized to receive real-time biofeedback from the VE (intervention) or shown still images (control). Primary outcomes were: (i) distance walked at baseline compared to the final 5 min cycle of week 2; (ii) the Lower Extremity Functional Index (LEFI) questionnaire. A satisfaction questionnaire was used. Follow up was to 24 weeks. Results: Total distance walked was significantly higher in the intervention group (*p* < 0.05), and 33% (2/6) of the intervention group had a clinically important LEFI improvement compared to 0/4 in the control group at week 2. No secondary outcome measures demonstrated any significant differences. The intervention received high satisfaction scores, significantly greater than the control group at week 24. No harms were recorded. Discussion: This feasibility study showed that VE and treadmill-walking improved walking distances and function for subjects with motor neglect. This is a promising novel approach and requires further validation through larger study.

## 1. Introduction

Motor neglect is a potentially reversible condition that describes a loss of function without a loss of strength, reflexes, or sensation [1]. Chronic pain conditions such as low back pain (LBP), fibromyalgia (FMS), and Complex Regional Pain Syndrome (CRPS) affect hundreds of thousands of patients in the UK each year and have large economic and health costs [2,3]. These conditions share the common feature of motor neglect or neglect-like symptoms with a presumed mechanism of learned non-use [4]. Weakness and loss of function have been described in CRPS patients with associated electrophysiological and cortical changes [5,6]. These changes have been associated with decreased motor control [7]. Intensive rehabilitation is the current gold standard for such patients; for example, randomized controlled trials recruiting patients with CRPS mirror therapy; motor imagery; desensitization and graded return to activities improve function [8,9]. Cortical changes can also be seen to improve following physical therapies [6].

Virtual Environments can be integrated into rehabilitation programmes to promote sensorimotor re-education, optimize function and improve engagement [10,11,12]. Attention-based repetitive training for functional tasks have been demonstrated to improve outcome [13]. Repetition, feedback, and motivation are beneficial for optimal outcome [13,14]. In one small study, Sato [15] used visual feedback in the form of virtual mirror therapy for treating five patients with CRPS, and found a more than 50% pain reduction in four of the five patients. In another study, six patients with CRPS engaged in functional tasks in a virtual kitchen [16], four of the six patients in this study reported subjective improvements in their pain and function, although objective pain measurements did not show any clinically meaningful improvements. These studies show some support for the use of real-time, visual biofeedback for the upper limbs. There have been no studies that investigate the use of visual biofeedback for patients with motor neglect affecting their lower limbs.

Integrating virtual environments and real-time biofeedback for gait training during rehabilitation is in the early stages of development [17,18]. Experimental systems that combine virtual environments with treadmills were first developed to investigate mechanisms of natural locomotion [19,20] including the close coupling observed in visuo-locomotor interactions in conscious [21] and sub-conscious [22] visual processing. Avatar-based visual biofeedback has mainly been clinical studies in children with cerebral palsy [23,24]. No studies have integrated virtual environments with real-time, visual biofeedback for gait training in patients with chronic pain, or with motor neglect affecting their lower limbs.

Although terms such as “immersion” and “presence” are used interchangeably even in professional VR literature, the former is best defined as the objective (technological) level of stimulus sensory fidelity, while the latter as its subjective effects or the response of the user [25,26]. A key factor for immersion is visual field size, measured as the subtended visual angle at the nodal point of an observer’s eye (whether monocularly or binocularly), which for fixed-sized stand-alone displays in a room depends on viewing distance while for head-mounted displays (HMDs) depends on display size alone. Unlike stand-alone displays, HMDs also remove any view of the user’s surroundings, potentially raising safety concerns when used in a clinical context. Other immersion factors include user tracking, such using an embodied representation of the user as an avatar, and display fidelity, including visual resolution, frame rate, graphical rendering, stereoscopy, audio quality, etc. Recent studies have shown how immersive technologies can enhance realism [27] or used in clinically meaningful applications [28]. In a recent review, aggregating 115 effects from 83 studies, technological immersion factors were found to have a medium-sized effect on user presence, while user tracking, visual field size, and stereoscopy were most impactful [26]. In relation to user tracking, it is uncertain as to what level of embodiment is needed to translate to a clinically meaningful benefit. Illusory ownership of a virtual body can be achieved in both first and third person perspectives under congruent visual and sensorimotor conditions [29].

Virtual systems take advantage of innovative technology to provide remote medicine opportunities. These are cost-effective in reducing therapist time which is the single largest cost for intensive rehabilitation strategies. Automating patient-preferred virtual rehabilitation will undoubtedly yield health savings, which are likely to be significant given the burden of economic costs associated with these conditions.

We designed a randomized controlled trial to assess the feasibility of a virtual environment system for rehabilitating patients with motor neglect. We aimed to investigate whether a virtual environment system in conjunction with a treadmill was feasible for the rehabilitation of subjects with motor neglect due to chronic pain conditions. In addition, we aimed to investigate whether a virtual environment system in conjunction with a treadmill can improve outcomes such as distance walked, pain, function, emotional well-being, and activity.

## 2. Materials and Methods

Patients were recruited from outpatient clinics and specialised rehabilitation programmes for chronic pain conditions. Potential subjects were identified by the treating consultants and therapists if they met both of the following criteria: (1) a chronic pain diagnosis of more than 3 months duration (e.g., sciatica; complex regional pain syndrome, fibromyalgia); (2) motor neglect as assessed by standard clinical examination by a physiotherapist trained to detect such motor neglect. No patient had suffered any brain lesion. The exclusion criteria were (1) patients with active serious medical problems that might affect their ability to participate in the exercise protocol (e.g., ongoing sepsis; recent myocardial infarction); (2) patients who are unable to use treadmill safely as judged by the screening physiotherapist; (3) patients who are unable to give informed consent, either through issues relating to competency or to language; and (4) patients with significant previous experience of virtual reality rehabilitation. Recruitment for the study was between May 2017 to May 2019. The study protocol was approved by the Cambridge University Hospitals NHS Foundation Trust Local Ethics Committee (17/LO/0299). The trial was registered on Clinicaltrials.gov (NCT03887962) and the full protocol can be viewed on www.clinicaltrials.gov. Written informed consent was obtained from all subjects. The trial has been reported according to CONSORT checklists, with extensions for feasibility and non-pharmacological trial reporting [30,31].

### 2.1. Intervention

Within two weeks of being successfully screened, subjects were randomized to either the study or control group in a 1:1 ratio using a sequential concealed allocation, which was pre-determined prior to study recruitment by a random number generator and held secure with an individual independent from the study group.

The study group was instructed to walk on a treadmill, following a “virtual path” displayed on a flat screen in front of them. A software-modified Kinect v2^TM^ device projected the subject’s avatar in a rendered environment displayed on a large screen in front of the treadmill. The environment was the same for all subjects in this group. The display, a 54 inch LED monitor placed in portrait mode, rendered a virtual humanoid avatar which mimics the movements of the subject in real time, emulating a third-person real time biofeedback. The visual angle subtended by the displayed stimulus at the nodal point of the eye, based on a viewing distance of 1.5 m between the display and a typical position of the patient on the treadmill, is approximately 12.6 deg by 21.6 deg. The control group were instructed to walk on a treadmill, while observing the flat screen in front of them, that displayed random scenes from the virtual reality environment thus controlling for attentional and non-movement related clues. A schematic of the setup is shown in Figure 1.

Subjects in both groups underwent rehabilitation using a treadmill for 3 times a week for two weeks. This treadmill rehabilitation was undertaken at the Gait Laboratory of Addenbrooke’s Hospital, Cambridge University Hospitals NHS Foundation Trust.

Walking on the treadmill was explained and demonstrated, and the subjects had time to practice this task during the screening visit. For extra safety, the subjects were allowed to use the handlebars of the treadmill for extra support if required. At each session, subjects were required to complete five walking cycles of up to five minutes, with a 3 min break in between cycles. The length of the walking cycles was determined by the subject. Every 30 s, the supervising therapist asked whether the subject would like to change the speed of the treadmill. Each subsequent cycle started at the speed at which the previous cycle finished. Each session commenced at the baseline speed and increased or decreased as outlined above.

### 2.2. Outcome Measures

Subjects completed questionnaires for subjective outcomes whilst the treadmill and Kinect v2^TM^ device collected objective outcome measures. These were collected at baseline (week 0) and the last visit of week 2. Questionnaires were also collected by post at week 24.

The primary outcomes for the study were functional improvement measured by: (1) the distance walked in the first 5 min cycle of the session at baseline, compared to the final 5 min cycle of week 2; and (2) the Lower Extremity Functional Index (LEFI) questionnaire, expressed as the percentage achieving MCID (Minimal Clinically Important Difference) change score of 9.

The following standardized, patient-reported outcome measures were completed at each time point and used as secondary outcome measures: (1) Brief Pain Inventory (BPI) for measuring pain, (2) Human Activity Profile scores, (3) Hospital Anxiety and Depression (HAD) scale to measure anxiety and depression; and (4) Neglect-like Symptoms Questionnaire (NLSQ) to measure depersonalization. A subject satisfaction questionnaire using Likert scales assessed the overall experience.

In addition to the walking distance, the machine recorded movement data on stride length and gait cycle (symmetry, stance, length), allowing the analyses and summaries of objective outcomes. This was downloaded in an anonymized report as a digital spreadsheet and analysed remotely. Part of the feasibility trial was to test the acceptability and reliability of such data capture in real time, using therapists who had no previous exposure to such technology.

### 2.3. Data Analysis

The data was assessed for its parametric/non-parametric properties and appropriate statistical analysis was performed. T-tests were used for parametric data. Wilcoxon–Mann–Whitney test was used for non-parametric data. The data has been presented without confidence intervals due to the small numbers in each group. As a feasibility study no power calculations for sample size were made.

## 3. Results

The flow diagram (Figure 2) outlines the recruitment to the trial. A total of 19 potential subjects were approached, 2 of whom did not feel that they could commit to the study. Of the 17 screened subjects, 7 could not arrange a time to attend for their treadmill sessions within the recruitment period. The 10 remaining subjects were randomized. All completed their treadmill sessions and questionnaires. Two subjects (both from the intervention group) did not complete their 24 week follow up questionnaires. A pragmatic decision based on funding and therapy resources necessitated a termination of the study in May 2019. Only returned data was included in the analysis with the 24 week questionnaires.

Table 1 outlines the baseline demographics and clinical characteristics of each of the groups. The groups were all female and whilst females are overrepresented in populations with chronic pain diagnoses, it was unexpected to not recruit any males. Seven out of the nine subjects who were not randomized were females who were slightly younger and were more likely to have a diagnosis of fibromyalgia.

The two groups were well matched with regards to their baseline outcome measures and there were no significant differences. The control group had slightly less pain (5.3 vs 6.0) and were taking fewer pain medications (0.7 vs. 2.2). They also had a trend to more anxiety (10 vs. 7.8) but again this was non-significant. There were significantly higher self-reported activities on the HAP maximally (73.3 vs. 61.3) and adjusted (49.3 vs. 34.5). This was corroborated by the higher baseline distance walked (151.2 m vs. 106.1 m) suggesting that the control group was more active than the intervention group at baseline.

One subject (intervention group) reported more pain and delayed a session of their treadmill as a result. This was not thought to be related to the intervention but rather the use of a treadmill. No other adverse effects were reported by any of the subjects including no disorientation nor dizziness as a result of using the virtual environment technology.

### 3.1. Primary Outcomes

The first a priori primary outcome measure was met. The total distance walked on the treadmill for the last 5 min of the last session when compared to the first five minutes of the first session was significantly higher in the intervention group (see Figure 3). All six subjects were seen to improve their distance walked compared to 2/4 from the control group. This was significant to *p* < 0.05 when analysed using the Wilcoxon–Mann–Whitney test. Secondary post-hoc analyses (Appendix A) confirmed that there was no intra-session bias to account for this. The mean total distance walked at baseline in the control group was 151.2 metres and this dropped to 139.6 metres at week 2. In the intervention group, however, the mean total distance walked at baseline was 106.1 m which increased to 141 m.

The second a priori primary outcome measure was also met. One out of six in the intervention group had an improvement in their self-reported LEFI questionnaire greater than the minimally clinically important difference (MCID) compared to 2/4 in the control group (see Table 2). Of the 4/6 in the intervention group who did not have an improvement of nine points, three had non-significant gains and one dropped by a non-significant four points. This translates to a number-needed-to-treat of six to improve self-reported lower limb function, although due to the small numbers in each group, this has wide confidence intervals that are not significant. The self-reported LEFI was then seen to return to baseline in both groups at week 24 suggesting that longer term gains in functional improvement were not sustained from this short but intensive intervention.

### 3.2. Secondary Outcomes

These are listed in Table 2. The average pain score out of 10 appeared to slightly reduce in the intervention group when week 2 (5.8) was compared to baseline (6.0). This was a sustained but non-significant reduction with the week 24 average pain score (5.75). Average pain scores increased slightly in the control group over the trial (5.3) at baseline compared to 6.5 and 6 at weeks 2 and 24, respectively. It should be noted that the intervention group was taking more medication at baseline (mean = 2.2) compared to the control group (0.7). The number of medications taken were seen to decrease in the intervention group at week 24 (0.75), whereas the control group continued to take the same amount of medication (0.75). Further evidence that the intervention is likely to be analgesic was seen in all the secondary outcomes which showed small and non-significant improvements including anxiety, depression, and activity. This compares favorably to the control group which saw decreases in self-reported activity and increases in anxiety and depression at week 2. Further evidence that the intervention was well received and beneficial to the subjects was the self-reported satisfaction questionnaire which showed a non-significant response, albeit favourable to the intervention at week 2 but was then seen to be significant at week 24. Subjects in the intervention group were very supportive of the technology. Taken together, the secondary outcomes support the intervention as an effective analgesic and rehabilitation that requires further investigation.

The mean total distance walked was significantly different when the week 2 scores were compared to baseline. Across all of the trials in the final session of week 2, the control group walked less (139.6 m) than when compared to the first baseline session (151.2 m). The intervention group however walked significantly more at week 2 (141.0 m) compared to baseline (106.1 m). The technology also collected gait data including stride length and swing-stance ratio [17]. There were no significant differences between the groups when this data was analysed. It was reassuring to see that this data was reliably captured and was able to be analysed remotely.

## 4. Discussion

The feasibility trial study of a virtual environment to rehabilitate patients who demonstrate motor neglect demonstrated that the technology was well tolerated and safe for the small numbers of subjects who were randomized to this arm. It showed that there were good functional improvements to be gained in the intervention group compared to the control group in terms of walking distance. This association was reinforced by the secondary post hoc analysis. Whilst the numbers in the study were small, this is encouraging. The functional gains were also seen subjectively with 1/6 of the intervention group achieving a MCID improvement than nine points on the LEFI. None of the control group did. Using this data and the LEFI as a primary outcome, a sample size of 50 subjects would need to be recruited to achieve a power of 0.81 in a larger study. Due to the small numbers of research participants, no conclusive statements can be given with regards to the analgesic and rehabilitation efficacy but there appeared to be improvement in all self-reported measures when compared to the control group which did not show similar improvements.

### 4.1. Context within Other Studies

This study replicates the beneficial effects of virtual environments in being able to rehabilitate patients who have neurological and musculoskeletal deficits.

Results from several systematic reviews show that using virtual reality to improve walking function in patients with neurological conditions, including walking speed, balance, and mobility [10,32]. The latter review found that virtual reality-enhanced gait training was more effective than identical training without virtual reality, to improve spatiotemporal gait parameters, including walking speed, in post-stroke patients. We found similar outcomes in our patient population.

A recent systematic review on virtual reality for spinal pain [33] mainly focused on the effect of virtual reality on pain and function as measured by self-reported outcome measures. In the studies that focused on chronic low back pain, the results showed a clinically important difference for pain intensity at short-term follow-up. Only one study looked at the effectiveness of virtual walking on pain and function [34]. This small study examined the effectiveness of adding passive virtual walking, where patients were asked to watch a virtual walking video clip in sitting position, to a short, intensive course of physiotherapy. The experimental group in this study showed a significant improvement in the distance walked during the six-minute walk test at the end of the test period. There was also a significant reduction in pain at the end of the intervention compared to the control group (visual analogue scale from 6.00 ± 1.06 to 2.52 ± 1.80 points). The study did not include any long-term follow-up. The study included 10 sessions over two weeks and combined it with intensive physiotherapy treatment. They did not evaluate the effect of virtual walking alone. Our study found similar increases in function (walking distance) using the virtual environment combined with walking alone, in six sessions over two weeks.

Hennesy et al. [35] undertook a small feasibility trial testing a virtual reality app during walking and reaching for patients with chronic low back pain. In this graded exposure approach to the functional task, they found that patients were able to undertake more difficult tasks over the sessions, measured by rate of perceived exertion, without increasing their pain or fear of movement. We found that patients with motor neglect were able to gain good functional improvements without increasing their pain level. These improvements were not maintained at long-term follow-up; this could be that the dose for the intervention tested in this study was not set at optimal level.

A previous study found that using a simplified avatar produced by eight markers, provided biofeedback on gait parameters during gait training [23]. Our study uses real-time, visual biofeedback from a full-body avatar from 25 markers that provides the subjects with feedback about the quality and accuracy of their movement.

The use of an avatar creates a third-person perspective, where a person sees himself or herself in the virtual environment as if from an observer’s viewpoint [36]. The third-person perspective is common in less-immersive virtual environments using motion-controlled virtual bodies, such as gaming [29]. It has been recommended for training where feedback about posture is important [37]. 

We hypothesize that the visual feedback from seeing their avatar move in the virtual environment helps to produce change through interoceptive neuroplastic mechanisms.

To date, most studies exploring virtual environments for chronic pain, including back pain and CRPS, have focused on pain reduction as the primary outcome. Most studies testing virtual reality as treatment for CRPS focused on the upper limb [15,16,38]. To our knowledge, our study is the first to include lower limb motor neglect in subjects with different chronic pain conditions, and to use function (walking distance) as a primary outcome. Currently there is no cure for chronic pain, and rehabilitation for this condition focuses on helping patients with self-management of this long-term condition and improving their function. Using virtual environments might help patients achieve an improvement in their walking distance, without increasing their pain. This small study shows promising results and further, high-powered studies to evaluate efficacy are indicated, including dose-finding studies (duration of treatment) to ensure that functional gains are maintained in the long-term.

In a recent review on the effects of virtual reality on pain, Trost et al. [39] proposed a heuristic model that summarizes the user experiences and key mechanism of action that affects the outcome of virtual reality on the pain experience. We propose that the additional feedback from the motion-controlled avatar in our study, provided a level of interactivity that leads to functional changes in people with motor neglect related to chronic pain conditions. Less immersive virtual environments have been recommended where the goal of treatment is functional restoration [40], and is considered the safest option when treatment includes ambulation [41]. Our study confirms that less-immersive virtual environments are sufficient to lead to functional changes in patients with chronic pain conditions, even though their pain did not change.

### 4.2. Weakness

This is a single center study. It would be important to see if these results can be replicated in other centers. The recruitment to the trial was low reflecting the challenging protocol with a limited time between screening and delivering the treadmill rehabilitation. Accessibility to the treadmill was limited to secondary care which reduced the number of subjects who felt that they could attend six times within a fortnight. Nine (45%) of the subjects who were approached and who expressed an interest in contributing to the study were not randomized as a result. Another weakness in the study design is that the subjects could not be blinded. We do not believe that this would significantly impact upon the results given that the objective and subjective measures both reported similar improvements. However it should be recognized that the outcome measures may have been influenced by this and a larger multicenter study is needed to confirm these findings. The intervention’s impact was thought to be mitigated by offering the control group an attentional control. This was delivered using a fixed image on the monitor at which subjects in the control group were asked to look. The use of such a control would reduce any efficacy difference that the intervention may have. Many patients who have chronic pain syndromes do not have motor neglect. It remains to be seen how generalizable the technology will be to the majority of patients with chronic pain syndromes. However, it is promising that the intervention was well tolerated, and subject satisfaction was high which suggests that this will be a generalizable technology.

The use of a less immersive technology is interesting and is a possible weakness that can be further explored.

### 4.3. Further Work

This feasibility study has demonstrated that intervention technology was easy to set up, reliable to operate and captured valid data that could be analysed remotely. Subjects reported no significant side effects from either the treadmill or the technology. Therapists fed back favorably with regards to the operational aspects of collecting data and delivering the intervention. The challenges to recruitment were largely due to access and consideration should be made in regards to protocol development to offer a greater gap between screening and initiating the treadmill sessions as well as providing more accessible locations, including community settings rather than secondary care. A larger multi-center study is indicated given the favourable results of this feasibility study that should take into account the changes in protocol and the accessibility challenges seen.

It is possible that more immersive devices may lead to greater benefits and this could be another fruitful area of further research.

## 5. Conclusions

Subjects with chronic pain syndromes and lower limb motor neglect were recruited to the intervention group in this single center, single-blinded feasibility study, and completed two-weeks of intensive treadmill-based rehabilitation with a virtual environment and real-time biofeedback through a virtual avatar representation. 

The intervention group demonstrated significantly further walking distances and self-reported function of their lower limb at two weeks when compared to the control group who received an attentional cue on the same treadmill. 

The potential benefit from this feasibility study that this Virtual Environment could offer patients who have motor neglect is clear and a further trial is indicated which would be best delivered with more subjects (at least 50) in a multi-center setting.

## Figures and Tables

**Figure 1 brainsci-11-00464-f001:**
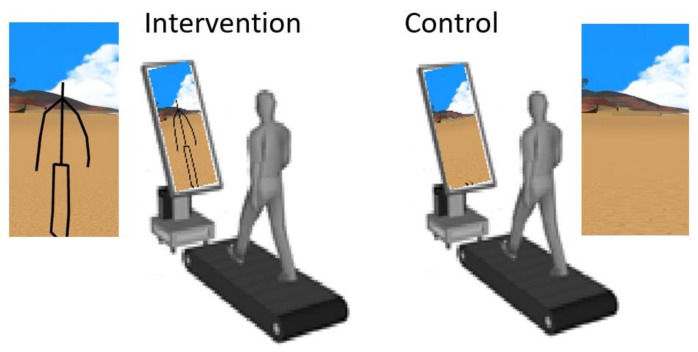
Schematic of the setup (incl. hardware and software) for capture of gait data during the trials. The setup includes a treadmill, screen (in portrait mode), the Kinect v2™ sensor (placed before the screen), an attached computer and associated software required for the capture.

**Figure 2 brainsci-11-00464-f002:**
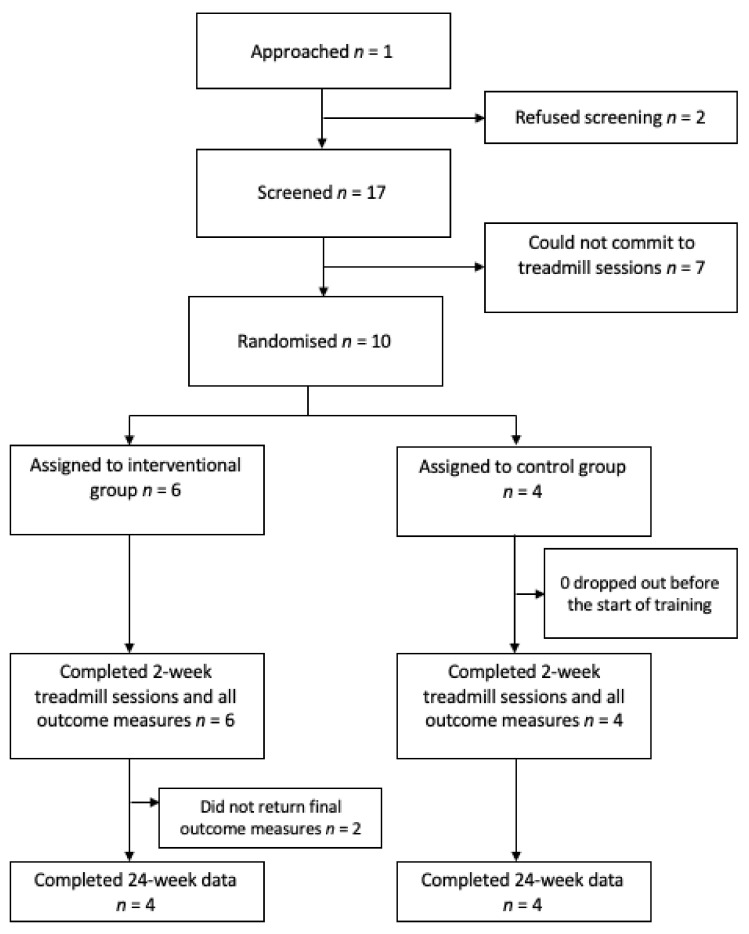
Flowchart of Virtual Environment Rehabilitation for Patients with Motor Neglect Trial (VERMONT) recruitment.

**Figure 3 brainsci-11-00464-f003:**
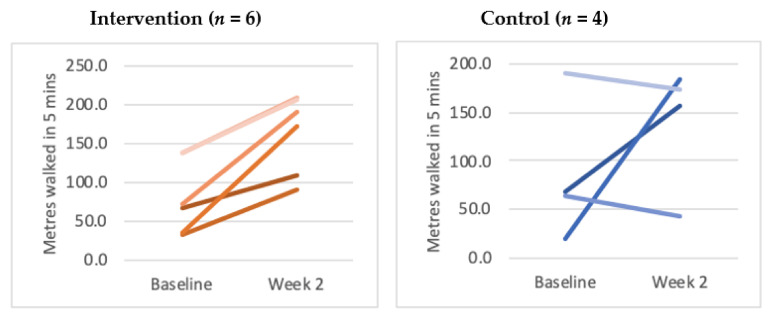
Distance walked in metres at baseline compared to week 2. (Please note: two of the lines in the intervention group overlap).

**Table 1 brainsci-11-00464-t001:** Baseline demographics.

	Intervention	Control	Not Randomized
N	6	4	9
Age (years)	55.2	52.0	48.3
Female (%)	6 (100)	4 (100)	7 (77%)
Diagnosis			
Sciatica	3	2	2
CRPS	3	1	4
FMS	0	1	3
Comorbidities			
Chronic pain syndromes	5	4	
Osteoarthritis	3	2	
Other *	6	5	
Depression	0	1	

* Other conditions included Asthma, hypertension, fracture, gastric reflux, rectal bleed.

**Table 2 brainsci-11-00464-t002:** Primary and Secondary Outcomes from VERMONT.

	Intervention Baseline	Control Baseline	Intervention Week 2	Control Week 2	Intervention Week 24	Control Week 24
**Primary Outcomes**						
Total distance walked in 5 min (m)	86.7	85.7	166.1 (91.6%) *	139.6 (62.9%) *	–	–
LEFI (/80) Raw scores Difference No. MCID > 9 pts	25.2–	38.3–	30.2 + 5.0 2/6	31.3 –7.0 0/4	24.5 –0.7 0/4	38.9 + 0.60/4
**Secondary Outcomes**						
*Questionnaires*						
Mean paint/10	6.0	5.3	5.8	6.5	5.75	6
Mean number of medications Reduced meds (*n*)	2.2	0.7	2.5 2/6	1.5 0/4	0.75 3/4	0.75 3/4
Mean Paint/10 interference scores	6.6	6.5	5.9	6.5	6.14	7.14
NLSQ/6	2.6	3.2	2.2	2.7	2.8	3.4
HAD (Anx)/21	7.8	10	6.5	12.3	8.3	12
HAD (Dep)/21	8.7	8.3	8.2	9.0	8.3	9.5
HAP (Max)/94	61.3 *	73.3 *	62.5	67.0	65.0	78.5
HAP (Adj)/94	34.5 *	49.3 *	38.5	43.7	41.0	40.3
Satisfaction/5	–	–	3.7	3.3	5 *	3.5 *
**Machine Outcomes**						
Mean distance walked (meters)	106.1	151.2	141.0 *	139.6 *	–	–
Swing stance ratio	0.34	0.40	0.34	0.39	–	–
Single double ratio	0.33	0.25	0.32	0.27	–	–
Absolute left right swing stance ratio	0.0139	0.0185	0.0113	0.00488	–	–
Absolute left right single double ratio	0.00249	0.0200	0.00032	0.00704	–	–

* Intervention and control groups significantly different *p* < 0.05 using Wilcoxon–Mann–Whitney. *LEFI* Lower Extremity Functional Index; *MCID* Minimal Clinically Important Difference; *NLSQ* Neglect-Like Syndrome Questionnaire; *HAD* Hospital Anxiety and Depression Scale; *Anx* Anxiety; *Dep* Depression; *HAP* Human Activity Profile; *Max* Maximal score; *Adj* Adjusted Score.

## Data Availability

The data are available from the corresponding author upon request.

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
