# Peer review of "Virtual Environment Rehabilitation for Patients with Motor Neglect Trial (VERMONT): A Single-Center Randomized Controlled Feasibility Trial"

_brainsci, 2021, doi:10.3390/brainsci11040464_

Round 1
Reviewer 1 Report
Summary
In the manuscript entitled “Virtual Environment Rehabilitation for Patients with Motor Neglect Trial (VERMONT): A Single-centre Randomised Controlled feasibility trial” the authors have conducted a pilot study using virtual reality feedback during treadmill walking. They state that this treatment improved walking and self-reported LEFI compared to the control group.
A. MAJOR ISSUES.
The topic of this work is of relevance and interest. Nevertheless, the work has a number of weaknesses that should be addressed by authors.
- Introduction.
I suggest the authors provide a more exhaustive description of motor neglect. The authors refer to Ramachandran and Altschuler (2009), but this paper is about the use of mirror visual feedback in some neurological conditions and does not specifically address motor neglect.
The authors state (lines 38-41) “Chronic pain conditions such as low back pain (LBP), fibromyalgia (FMS) and Complex Regional Pain Syndrome (CRPS) … these conditions share the common feature of motor neglect [4].” However, Acerra et al. [4] specifically address CRPS, stroke and phantom limb. The authors should include references appropriate with the text.
- Materials and Methods.
Motor neglect can result from different brain lesions, however the authors do not indicate if their patients had suffered any brain lesion. They should indicate if the participants had suffered brain damage and provide detailed information about this. Also, they should inform if the participants also suffer sensory neglect and which side was affected by motor neglect.
- Results.
Missing data shouldn’t be included in the 24-week follow up. It is not clear why these participants dropped out from the study, therefore including them as an average of other participants that completed the follow up is a serious bias. The authors should analyse the follow up just taking into account the 4 participants of the intervention group that completed the questionnaires.
3.1. Primary outcomes.
The authors just compared the performance during the initial 5 minutes of the first session with the final 5 minutes of the last session. This doesn’t provide control over a possible intrasession effect. I mean, the participant could show a worse performance at the beginning of every session and improve as the session progresses. The analysis of the walking distance should include all the sessions as a repeated measures analysis with sessions as within subject factor and group as between subject factor. Furthermore, the authors could also analyze possible differences within every session by comparing the first and the last 5 minutes.
The figure 3 should show the progression of participants along all the sessions, not just the initial and final 5 minutes of the first and last session. The distance walked in the follow up should be also included in the figure 3.
Line 198. The authors mention that 2/6 of the intervention group had an improvement in their self-reported LEFS questionnaire. How did respond the other 4 participants? just <9 points or did worse than the baseline?
3.2. Secondary outcomes.
Follow up pain scores shouldn’t include participants that did not complete the questionnaires.
Table 2 shows that intervention group scored 5 out of 5 in satisfaction during the 24 weeks follow up. Could the authors give any reason for this?
- Discussion.
The authors state that there is a significant functional improvement in terms of walking distance. However, as I mentioned above their analysis just compared two short periods of time.
The authors point out differences in subjective scales (LEFS and HAP) between intervention and control groups. Nevertheless, these results must be taken very cautiously as the participants were not blind to the treatment. Moreover, 45% of participants declared to be interested in contributing to the study. Even if they were randomly allocated to each group if the information sheet gave them information about the intervention this could have influenced their responses to the scales.
In this work participants movement was captured through a Kinect sensor and translated into an avatar displayed on the TV screen for the treatment group. The authors mention: “A previous study found that using a simplified avatar produced by 8 markers, provided biofeedback on gait parameters during gait training [22]. Our study uses real-time, visual biofeedback from a full-body avatar from 25 markers.” The authors should discuss the potential effect of the feedback from the avatar in comparison with watching a simulation of movement (first person visual flow) during the walk, as it happens in real life. They should also discuss deeply why this visual feedback produces such improvement compared to just walking watching a still photo. Which physiological mechanisms support this effect? The authors should provide some hypothesis about brain plasticity mechanisms, cortical rearrangement, compensation…
B. MINOR ISSUES.
Line 144. “Human Activity Profile” should include (HAP) acronym immediately after.
Figure 2. In the control group arm it says “X dropped out…”, it should say “0 dropped out…”.
Line 173. I guess the authors mean “this was not a surprise”.
It is not clear if there is some supplementary material available. If there is any it should be available.
Reviewer 2 Report
The manuscript reports the results of an experimental investigation focused on the feasibility of a virtual environment for patients with motor neglect and chronic pain symptoms. It was found that VE as a study condition and treatment method improved walking distances and function in a treadmill task (3 times per week for two weeks, with 10 participants).
The manuscript is well structured and written, and the study is presented in a transparent way. The study is based on a specific group of participants with a difficult level of acquisition. The results are promising toward the implementation of an innovative and virtual treatment method. It would be interesting to see whether the promising results could replicated based on a larger and more robust group of participants.
There are three points which I would like to comment on and which the authors should consider in a revised version of a manuscript:
- As usual in papers based on clinical studies, the paper is very compact and focused on a clear research question. What I miss in the background section is a bit more detailed view on the topic of virtual environments. One important user- and cognition-oriented aspect in the ongoing debates on virtual environments is immersion. This aspect is discussed in some of the papers you cite but not a single time in your manuscript. The VE application used in this study is based on a large screen in front of treadmill. If participants had been confronted with a smaller screen or even modern VR headset (OculusRift or HTC Vive), the results could have been influenced considerably due to different levels and possibilities of immersion. Could you integrate immersion as an aspect in your weakness (4.2) or further work chapter (4.3.)? It would also be important that you introduce virtual environments in more detail at the beginning of your paper and show current developments of the possibilities to present these spatial data and information. I would recommend two quite recent papers. J. Keil and colleagues, among other aspects of spatial data representation, point to the facts that different visual fields in modern VR systems influence perception of the represented environment. In a cognition-oriented study with older adults, I. Lokka and colleagues show that different forms of realism in an immersive virtual environment influences route navigation performance:
Keil, J., Edler, D., Schmitt, T., Dickmann, F. (2021): Creating Immersive Virtual Environments Based on Open Geospatial Data and Game Engines. In: KN - Journal of Cartography and Geographic Information, 71, online first: https://doi.org/10.1007/s42489-020-00069-6
Lokka, I.E., Çöltekin, A., Wiener, J. M., Fabrikant, S. I., Röcke, C. (2018). Virtual environments as memory training devices in navigational tasks for older adults. Scientific Reports, 8(1). https://doi.org/10.1038/s41598-018-29029-x
- It would be nice to read in your concluding chapter which previous studies are extended by your results. This would clearly connect your valuable studies with other highlighted state-of-the-art literature.
- Is it planned to extend your study sample so that the robustness and replicability of these (interesting) findings of your feasibility study could be examined and reported? If yes, please point to your small sample size and your extended study in the future work chapter.
Round 2
Reviewer 1 Report
I appreciate authors’ efforts to reply my previous questions and to make the manuscript more clear for the reader. Nevertheless, I still have some concerns about this research.
My main concern is still about the analysis of the results. The authors compared the first 5 minutes of the baseline session with the last 5 minutes of the last training session. In the table S3 (I understand they show the raw data for each walking cycle of baseline and the last training sessions) they state in the revised text that there is not intra session bias. But they do not provide the statistical post hoc analysis supporting this statement. Just watching the table S3 I see there is an improvement in the walking distance as the session progresses both in control and intervention groups. Looking at cycle 0 for control and intervention, both at baseline and 2 weeks the results are almost identical, but at the end of the session there are differences 238.5 to 139.6 (control) and 127 to 163 (intervention), so there is an intra session bias. I do not know if differences are statistically significant. The authors mention that using data from the last 5 minutes of the baseline session would imply some level of rehabilitation. Rehabilitation is quantitative and progressive, so in case the intervention was effective it should show differences when sessions are compared. I recommend a repeated measures analysis, with session as within subject factor (including cycle as well for intra session effects) and group as between subject factor. The graph illustrating the effect of training should show the progression across all the sessions. In case there were significant differences among sessions it would permit to see at which time point training yields an effect. With the current analysis I do not know if there is effect of the training and it could be seen before the last session.
Regarding LEFI table showing raw data from 6 participants any of them show an increase above 9 points. In the text the authors state 2/6.
My doubt about the score 5/5 in satisfaction during the 24 weeks follow up is why the participants did not give this score immediately after using the technology but 24 weeks later. It is weird. Nevertheless, I do not consider it extremely relevant.
